# Lumenal calcification and microvasculopathy in fetuin-A-deficient mice lead to multiple organ morbidity

**Marietta Herrmann[1][�][¤], Anne Babler[1][�], Irina Moshkova[1], Felix Gremse[2], Fabian Kiessling[2], Ulrike Kusebauch[3], Valentin Nelea[4], Rafael Kramann[5], Robert L. Moritz[3], Marc D. McKee[4]\*, Willi Jahnen-Dechent[1]\***

**1** Helmholtz Institute for Biomedical Engineering, Biointerface Lab, RWTH Aachen University Hospital, Aachen, Germany, **2** Helmholtz Institute for Biomedical Engineering, Experimental Molecular Imaging, RWTH Aachen University Hospital, Aachen, Germany, **3** Institute for Systems Biology, Seattle, Washington, United States of America, **4** Faculty of Dentistry, Faculty of Medicine (Dept. of Anatomy and Cell Biology), McGill University, Montreal, Quebec, Canada, **5** Division of Nephrology, RWTH Aachen University Hospital, Aachen, Germany

উ These authors contributed equally to this work.
¤ Current address: IZKF Group Tissue Regeneration in Musculoskeletal Diseases, University Clinics and Orthopedic Center for Musculoskeletal Research, University of Wuerzburg, Wuerzburg, Germany
\* willi.jahnen@rwth-aachen.de (WJD); marc.mckee@mcgill.ca (MDM)

**Data Availability Statement:** All relevant data are within the manuscript and its Supporting Information files. Raw data of gene chip expression studies were submitted according to MIAME

## Abstract

The plasma protein fetuin-A mediates the formation of protein-mineral colloids known as calciprotein particles (CPP)–rapid clearance of these CPP by the reticuloendothelial system prevents errant mineral precipitation and therefore pathological mineralization (calcification). The mutant mouse strain D2,*Ahsg*-/- combines fetuin-A deficiency with the calcification-prone DBA/2 genetic background, having a particularly severe compound phenotype of microvascular and soft tissue calcification. Here we studied mechanisms leading to soft tissue calcification, organ damage and death in these mice. We analyzed mice longitudinally by echocardiography, X-ray-computed tomography, analytical electron microscopy, histology, mass spectrometry proteomics, and genome-wide microarray-based expression analyses of D2 wildtype and *Ahsg*-/- mice. Fetuin-A-deficient mice had calcified lesions in myocardium, lung, brown adipose tissue, reproductive organs, spleen, pancreas, kidney and the skin, associated with reduced growth, cardiac output and premature death. Importantly, early-stage calcified lesions presented in the lumen of the microvasculature suggesting precipitation of mineral containing complexes from the fluid phase of blood. Genome-wide expression analysis of calcified lesions and surrounding (not calcified) tissue, together with morphological observations, indicated that the calcification was not associated with osteochondrogenic cell differentiation, but rather with thrombosis and fibrosis. Collectively, these results demonstrate that soft tissue calcification can start by intravascular mineral deposition causing microvasculopathy, which impacts on growth, organ function and survival. Our study underscores the importance of fetuin-A and related systemic regulators of calcified matrix metabolism to prevent cardiovascular disease, especially in dysregulated mineral homeostasis.

format to gene expression omnibus GEO (GSE140954). Raw mass spectrometry data is deposited in PeptideAtlas at: http://www.peptideatlas.org/PASS/PASS00920.

**Funding:** This work was supported by grants awarded to WJD by the IZKF Aachen of the Medical Faculty of RWTH Aachen and by the German Research Foundation (DFG SFB/TRR219-Project C-03). This work was also funded in part by National Institutes of Health from the National Institute of General Medical Sciences (NIGMS) grants R01GM087221, S10RR027584 and 2P50GM076547 to the Center for Systems Biology, the National Science Foundation MRI grant 0923536, and the Canadian Institutes of Health Research. We thank Sarah Li and Lydia Malynowsky for expert technical assistance. MDM is a member of the FRQS Network for Oral and Bone Health Research.

**Competing interests:** The authors have declared that no competing interests exist.

## Introduction

High concentrations of extracellular phosphate are toxic to cells. Impaired urinary phosphate excretion increases the serum phosphate level and induces a premature-aging phenotype [1]. Patients suffering from chronic kidney disease (CKD)–especially patients on dialysis–have increased serum phosphate and associated high cardiovascular morbidity and mortality [2]. Recently we showed that the phosphate toxicity may in fact be caused by the uptake and inflammatory action of excess protein-mineral complexes called calciprotein particles (CPP) [3]. These particles form in supersaturated (with respect to hydroxyapatite) calcium and phosphate solutions also containing plasma proteins, especially the hepatic glycoprotein fetuin-A. The cellular clearance and biological activity of CPP depend on their maturation state and crystallinity. The particles undergo a typical two-stage ripening process, starting as spherical primary CPP measuring 50–150 nm in diameter, which are chemically unstable, contain amorphous mineral, and are preferentially cleared by endothelial cells [3]. Primary CPP spontaneously transform into oblongate secondary CPP ranging up to 500 nm in length; these are stable for at least 24 h at body temperature [4], they contain crystalline mineral and are therefore more rigid, and are preferentially cleared by macrophages of the mononuclear phagocytic system [5]. Colloidal CPP allow efficient transport and protected clearance of bulk mineral without risk of precipitation. However, fetuin-A is critically required to form stable CPP, and fetuin-A-deficient CPP are unstable and readily precipitate [6]. Similar to lipoproteins playing a critical role in lipid transport and metabolism, fetuin-A plays a critical role in solubilization and clearance from the circulation of calcium phosphate-protein complexes. In line with this, several studies have identified protein-mineral particles containing fetuin-A in granules formed by incubation of serum-containing cell culture media with calcium and phosphate [7], in the serum of bisphosphonate (etidronate)-treated rats [8], and in an animal model of renal failure [9]. Furthermore, the accumulation of fetuin-A-containing mineral particles was noted in the peritoneal dialysate of a patient suffering from calcifying peritonitis [10] and in patients suffering from chronic kidney disease (CKD) [11]. Recent work has shown that both the abundance of CPP [12–14] and the kinetics of CPP formation [15–18] in serum of CKD patients can be used to monitor the propensity for calcification of these patients. The function of fetuin-A as an inhibitor of calcification is underscored by clinical studies showing that low fetuin-A serum levels are associated with calcification disease [19–25].

Over some years now, we have generated fetuin-A deficient *Ahsg-/-* mice and backcrossed them onto two defined genetic backgrounds, C57BL/6 (B6) and DBA/2 (D2) [26, 27]. DBA/2 mice–like C3H, C3Hf and BALB/c mice–are prone to dystrophic cardiac calcification [27, 28], while C57BL/6 mice are relatively calcification-resistant. D2,*Ahsg-/-* mice are certainly one of the most calcification-susceptible mouse strains known [29]. Concurrent with this work we established that the calcification phenotype of D2,*Ahsg-/-* mice is governed by combined deficiency of fetuin-A, pyrophosphate and magnesium, thus at once affecting three potent extracellular regulators of mineralization [30]. Unsurprisingly, this prominent calcification phenotype is associated with bone abnormalities [31], decreased breeding performance, organ damage, and increased mortality. For example, severe renal calcinosis ultimately causes secondary hyperparathyroidism [29], and myocardial calcification is associated with fibrosis and diastolic dysfunction [32]. Despite the fact that D2,*Ahsg-/-* mice have one of the most severe phenotypes of ectopic calcification, the early events leading to this are poorly understood, because once started, calcification proceeds extremely rapidly.

Here, in the present study, we performed a detailed analysis of progressive worsening of the soft tissue calcification in D2,*Ahsg-/-* mice applying the techniques of computed tomography, histology, electron microscopy, genetic and proteomic analysis. We report that calcified lesions

first develop in these mice within the lumen of microvessels, indicating that calcification in these mice is caused primarily by disturbed mineral ion handling in the extracellular fluids including blood. One consequence of disturbed mineral matrix metabolism is myocardial calcification and dysfunction, which progressively worsens and is primarily responsible for premature death in D2,*Ahsg-/-* mice. These mice are well suited to study pathomechanism, diagnosis and therapy of cardiovascular consequences of dysregulated mineral homeostasis, a hallmark feature of chronic kidney disease CKD in the aging population [33].

## Material and methods

### Animals

Wildtype and fetuin-A-deficient mice on either the DBA/2N or C57BL/6N genetic background were maintained in a temperature-controlled room on a 12-hour light/dark cycle. Standard diet (Ssniff, Soest, Germany) and water were given *ad libitum*. Mice were kept at the animal facility of RWTH Aachen University Clinics. All animal experiments were conducted in agreement with German animal protection law and were approved by the state animal welfare committee.

The survival analysis shown in Fig 8 was carried out retrospectively based on routinely collected data in the period dating from 2007–2008. Breeding of DBA/2, Ahsg -/- homozygous mice was stopped after the higher death rate for fetuin-A deficieny in the DBA/2 background was realized. The strain is now maintained symptom-free as heterozygotes. Survival monitoring data were obtained on the daily basis from birth until natural death and documented in the institutional animal database as well as in records of laboratory personnel and qualified animal caretakers on corresponding animal ID cards. None of included animals had been euthanized. Trained and experienced animal caretakers and investigators have not detected any abnormalities in observed animals regarding their ability to move, access food and water, respiratory changes, body changes. Life span (age in days at the timepoint of natural death) data had been extracted from the total pool of the animals free of any kind of chemical or other interventions (e.g. used as control) separately for each genotype. Premature death was not accompanied with detectable suffering, animals stayed active and though they show slower movements they still reached food and water, had no wounds, nor respiratory difficulties.

At different ages (as indicated in the figures), mice were sacrificed with an overdose of isoflurane and exsanguinated. Animals were perfused with 20 ml ice-cold PBS to rinse blood from the circulation unless otherwise stated.

### Computed tomography (whole mice)

Mice were anaesthetized with isoflurane and placed in a high-resolution computer tomograph (Tomoscope DUO, CT-Imaging, Erlangen, Germany). The settings of the CT scan were: 65 kV / 0.5 mA, and 720 projections were acquired over 90 seconds each. Analysis of the CT scans was performed with the Imalytics Preclinical Software [34]. Full-body acquisitions were obtained to examine in three dimensions the overall state of calcification in whole mice.

### Micro-computed tomography (mouse organs)

Micro-computed tomography (Micro-CT, model 1072; Skyscan, Kontich, Belgium) of heart, lung, kidney, testis, skin, spleen and pancreas was performed to visualize entire calcification patterns in three dimensions and at higher resolution within various selected organs. The X-ray source was operated at a power of 40–50 kV and at 200–250 μA. Images were captured using a 12-bit, cooled, charge-coupled device camera (1024 x 1024 pixels) coupled to a fiber optic taper to the scintillator. Using a rotation step of 0.9˚, each sample was rotated through

180 degrees with a scan resolution of 10–18 microns per pixel. Data was processed and reconstructed using Skyscan tomography software TomoNT (ver. 3N.5), CT Analyser (CTAn ver. 1.10.0.2 and CT Volume (CTVol ver. 2.0.1.3).

## Histology and immunohistochemistry

For light microscope histology, organs were collected without perfusion and with immersion fixation in paraformaldehyde for at least 24 h. The fixed samples were subsequently dehydrated and embedded in paraffin. Hematoxylin/eosin and von Kossa staining for mineral was performed on 5-μm-thick sections. For immunohistochemical staining visualized by immunofluorescence microscopy, freshly dissected (unfixed) tissues were embedded in Tissue-Tek embedding medium (Sakura Finetek, Staufen, Germany), frozen in liquid nitrogen, and then kept at -20 ˚C until use. Cryosections were cut at 6-μm-thickness followed by fixation with Bouin's fixative. Calcified tissue was decalcified with 0.5 M EDTA overnight. PBS-washed sections were blocked with 5% goat serum (Dako, Hamburg, Germany) prior to primary antibody incubation. Primary antibodies were rat anti-mouse CD31 (BD Pharmingen, Franklin Lakes, NJ, USA, dilution 1:10), rabbit anti-mouse OPN (R&D Systems, McKinley Place NE, MN, USA, dilution 1:1000), rat anti-mouse OPN (Enzo Life Science, Lörrach, Germany, dilution 1:100) and rabbit anti-mouse Lyve-1 (Acris Antibodies GmbH, Herford, Germany, dilution 1:1000); all incubated for 1 hour. Primary antibodies were detected by secondary antibodies coupled with either Alexa Fluor 488 or Alexa Fluor 546 (all from Molecular Probes, Life Technologies, Carlsbad, CA, USA). DAPI (Sigma, Taufkirchen, Germany) was used as a nuclear stain. Stained sections were examined with a Leica DMRX fluorescence microscope (Leica Microsystems GmbH, Wetzlar, Germany) and DISKUS software (Carl H. Hilgers, Technisches Büro, Königswinter, Germany). Pictures were processed with Adobe Photoshop (Adobe Systems GmbH, München, Germany).

To detect early-stage calcified lesions, bovine fetuin-A (Sigma) was purified by gel filtration, and 100 μl of a 7 mg/ml protein solution was injected intraperitoneally one day before euthanasia. Fetuin-A was detected on cryosections using a rabbit polyclonal anti-fetuin-A antibody (AS 237) made in-house, followed by secondary Alexa Fluor 546 goat anti-rabbit antibody.

## Electron microscopy

For ultrastructural characterization by transmission electron microscopy (TEM), interscapular brown adipose tissue sections were fixed with 2% glutaraldehyde (Electron Microscopy Sciences, Hatfield, PA, USA) and immediately dehydrated through a series of graded ethanol dilutions. Samples were embedded in LR White acrylic resin or epoxy resin (Electron Microscopy Sciences). Ultrathin sections (80-nm-thick) cut with a Leica EM UC6 ultramicrotome (Leica Microsystems Canada, Ltd, Richmond Hill, ON) were placed on formvar-coated nickel grids (Electron Microscopy Sciences) and stained (or left unstained) with uranyl acetate and lead citrate (Electron Microscopy Sciences) for viewing by TEM. A field-emission FEI Tecnai 12 BioTwin TEM (FEI, Hillsboro, OR, USA) was used to image the stained sections at 120 kV.

Electron diffraction in the selected-area configuration (SAED) mode, and energy-dispersive X-ray spectroscopy (EDS), were performed on unstained sections at 200 kV using a Philips CM200 TEM equipped with a Gatan Ultrascan 1000 2k X 2 k CCD camera system model 895 and an EDAX Genesis EDS analysis system (FEI, Hillsboro, OR, USA).

## X-ray diffraction

X-ray diffraction (XRD) analysis was performed using a D8 Discover diffractometer (Bruker-AXS Inc., Madison, WI, USA) equipped with a copper X-ray tube (wavelength, 1.54056 Å), and

a HI-STAR general area detector diffraction system (Bruker-AXS Inc.). All components (X-ray source, sample stage, laser/video alignment/monitoring system and detector) were mounted on a vertical θ–θ goniometer. Measurements were run in coupled θ–θ scan in microbeam analysis mode (50 μm X-ray beam spot size). Samples used were tissue-embedded LR White acrylic blocks (the same block faces used for preparing the optical and electron microscopy sections) for small-area, localized analyses of mineral-rich regions identified on the block face.

## Gene expression analysis

Wildtype and *Ahsg*-/- DBA/2 mice at 6–7 weeks of age were used for gene expression analysis, using 2–3 female and 2–3 male mice. Kidneys and brown adipose tissue dissected from the kidney pelvic region were collected. Tissue samples were homogenized and stored in peq-GOLD RNAPure reagent (PEQLAB Biotechnologie GMBH, Erlangen, Germany). RNA extraction was performed using a standard phenol-chloroform extraction procedure or RNeasy Lipid Tissue Mini Kit (Qiagen, Hilden, Germany), respectively. The gene expression profile was analyzed with Affymetrix Mouse Genome 430 2.0 Arrays (Affymetrix, Santa Clara, CA, USA). Gene data analysis was carried out using Bioconductor [35] packages under R1. The quality of the microarrays was assessed using the ArrayQualityMetrics package [36]. The arrays were analyzed with the outlier detection algorithm within the package. Arrays with two or more outlier calls were considered of insufficient quality, and these were excluded from further normalization and data analysis. Background adjustment, normalization and summarization were applied using the RMA algorithm within the Affy package [35, 37]. Probe sets which could not be annotated to any gene, were removed from the expression set. A mean standard deviation of all probe sets was calculated. All probe sets which had a lower standard deviation than the mean standard deviation were excluded. Both filtering methods were performed using the Genefilter package [38]. Differential expression was probed using Bayesian statistics and Limma package [35]. Differences between genotype and sex, as well as a possible interaction of both, were calculated. Multiple testing correction was performed using the procedure of Benjamini Hochberg implemented in the Multtest package [39]. Probe sets with a p-value below 0.05 were considered as differentially expressed. Annotation of probe sets was applied with Annaffy, Annotate and Mouse4302.db packages [40–42]. Volcano plot representation was used to visualize differential expression. Differential expressed probe sets were tested for overrepresentation of KEGG gene sets. The analysis was performed using the GSEABase package [43] under Bioconductor. Raw data were submitted according to MIAME format to gene expression omnibus GEO (GSE140954).

## Protein analysis

Interscapular brown adipose tissue, skin (interscapular region) and heart were dissected. Calcified lesions were scraped out under a dissection microscope (Leica MZ6). Calcification-free tissue was collected as control tissue. Samples were snap-frozen in liquid nitrogen and stored at -70 ˚C. Samples were thawed, transferred to 2 mL reaction tubes, and incubated with SDS sample buffer (0.25 M TRIS, 8.2% SDS, 20% glycerin, 10% β-mercaptoethanol, bromophenol blue) containing 40 mM EDTA at 96 ˚C for 5 min at 10 μl per 1 mg tissue. The supernatant was removed, and tissue pellets were homogenized for 2.5 min at 25 Hz in a mixer mill (Tissue Lyser II, Qiagen, Hilden, Germany). Following this, samples were boiled for 5 minutes. Protein extracts were separated in 12.5% polyacrylamide gels using SDS-PAGE. Gels were washed and proteins were stained with Imperial Protein Stain (Thermo Fisher Scientific, Rockford, IL, USA) for 2 hours. Unbound stain was removed by washing in ultra-pure water overnight. Pictures of gels were recorded using a digital camera. Gel lanes were cut into 2-mm bands using a

GridCutter (The Gel Company, San Francisco, USA). Individual gel slices were subjected to in-gel reduction with dithiothreitol (10 mM, 30 min, 56 ˚C, Merck-Calbiochem, San Diego, USA), alkylation with iodoacetamide (50 mM, 30 min, room temperature, darkness, Fluka, St. Louis, USA) and digestion with 150 ng trypsin (5 h, 37 ˚C Promega, Sequencing Grade, Wisconsin, USA) using a Tecan Freedom EVO robotic liquid handler (Tecan Systems, Inc, San Jose, USA). Peptides were extracted with 50 µl 50% (v/v) acetonitrile, 50 mM ammonium bicarbonate, concentrated by centrifugal evaporation (Savant, Thermo-Fisher Scientific, USA) and re-solubilized in 20 µl 2% (v/v) acetonitrile, 0.1% formic acid, in water.

Peptides were analyzed on a LTQ Orbitrap Velos with nano electrospray ionization source (Thermo-Fisher Scientific, San Jose, CA) connected to a 1100 Series HPLC (Agilent Technologies, Santa Clara, CA) with an electronically controlled flow splitter for nano flow rates. Peptides were loaded to a trap column packed with ReproSil-Pur C18-AQ (15x0.075 mm I.D., 120 Å, 3 µm, Dr. Maisch, Ammerbuch-Entringen, Germany) for 6 min using 2% acetonitrile, 0.1% formic acid in water at a flow rate of 3 µl/min. Peptide separation was performed with an in-house packed capillary column (150x0.075 mm I.D., 120 Å, 3 µm, Dr. Maisch ReproSil-Pur C18-AQ, Ammerbuch-Entringen, Germany) using 0.1% formic acid in water (A) and 0.1% formic acid in acetonitrile (B) with a gradient from 2% to 35% B in 60 min at a flow rate of 0.3 µl/min. Survey full scan MS spectra were acquired in the mass range m/z 300–1800 in the Orbitrap analyzer at a resolution of 60,000. The five most intense ions in the survey scan were fragmented by collision-induced dissociation in the LTQ. Charge state one, and unassigned charges, were rejected. MS/MS spectra were acquired upon a minimal signal of 500 counts, with an isolation width of two and a normalized CE of 35. Dynamic exclusion was enabled to exclude precursors for 60 s after three observations. Raw mass spectrometry data is deposited in PeptideAtlas at: http://www.peptideatlas.org/PASS/PASS00920.

Thermo Xcalibur .RAW files were converted to mzML using ProteoWizard msconvert (version 3.0.03495) [44, 45]. MS/MS spectra were searched with Comet (version 2015.02 rev 5) [46] against the mouse proteome obtained from UniProt (www.uniprot.org, release 2016_05), common contaminants and a sequence-shuffled decoy counterpart were appended to the database. Peptides were allowed to be semi-tryptic with up to two internal cleavage sites. The search parameters included a fixed modification of +57.021464 to account for carbamidomethylated cysteines and variable modifications of +15.994915 for oxidized methionines and +42.010565 for N-terminal acetylation. The search results were processed with the Trans-Proteomic Pipeline (version 4.8.0 PHILAE) [47] including PeptideProphet [48], iProphet [49] and ProteinProphet [50]. Peptide spectrum matches (PSMs) generated by the search engine were analyzed with PeptideProphet to assign each PSM a probability of being correct. The accurate mass binning and nonparametric model were used in the PeptideProphet analysis. PeptideProphet results were further processed with iProphet to refine the PSM-level probabilities and compute peptide-level probabilities, and subsequently with ProteinProphet to calculate probabilities for protein identifications. Only proteins identified with a probability of $\geq 0.9$ in each sample corresponding to an error of $\leq 1\%$ were considered in this analysis. For further analysis, we focused on proteins, which were uniquely identified in calcified lesions but not in the surrounding intact tissue. Association of these proteins with functional pathways (KEGG) was derived by STRING (version 10, [51]) applying a minimum required interaction score of 0.9 (highest confidence).

## Results

### Widespread ectopic calcification in D2,Ahsg-/- mice

We performed X-ray-based computed tomography to survey ectopic calcification in the entire bodies of living D2 mice (Fig 1). Apart from the expected mineralization of the skeleton and

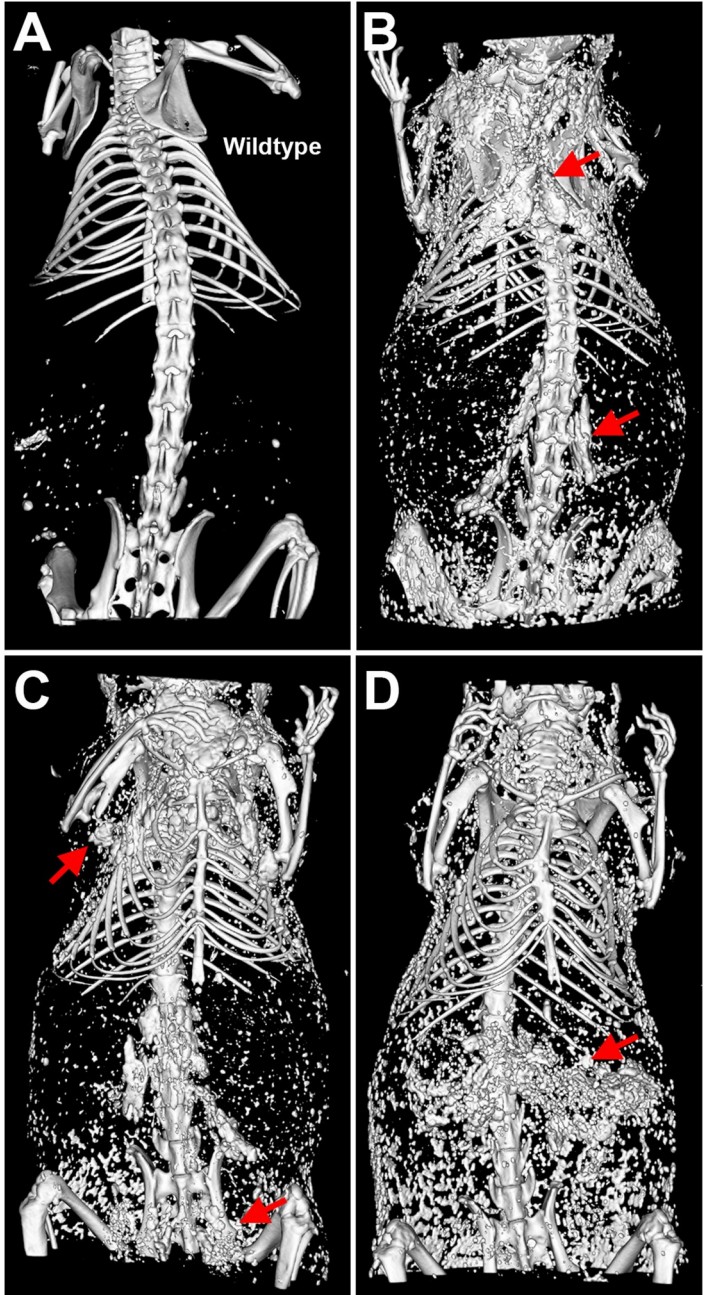

**Fig 1. Typical examples of adult calcified D2,*Ahsg*-/- mice.**

teeth, wildtype mice showed only minor amounts of mineral elsewhere in the gastrointestinal tract attributable to ingested dietary mineral (Fig 1A). In contrast, D2,*Ahsg*-/- mice presented with severe and abundant calcification spread throughout many soft tissues. We observed prominent mineral lesions in interscapular brown adipose tissue (BAT) (Fig 1B), surrounding the kidney and within the pelvis (Fig 1B), and in the axillae (Fig 1C). Computed tomography revealed calcified lesions in the spleen and pancreas (Fig 1D), testes (Fig 1C), kidney and skin (Fig 1B–1D, see also Supplemental Movies). Previously reported, mineral-containing lesions

in the lung and myocardium [29] remained undetected by computed tomography of live animals because of the motion disturbance caused by heartbeat and breathing movements; however, severe calcification was indeed detected in these two organs by post mortem microcomputed tomography of excised organs (Supplemental Movies). Supplemental S1A–S1H Fig show unstained tissues with nodular calcified lesions presenting as off-whitish, semi-transparent granules of sub-millimeter size. Nodules were present in the interscapular brown fat (A), adrenal fat (B), but not the kidney pelvis, of 6-week-old a D2,*Ahsg*-/- mice, as well as in the ventricular wall and the atrium (C), lung tissue (D), spleen (E), pancreas (F), kidney pelvis (G), and ovaries (H) of 16-week old D2,*Ahsg*-/- mice. In summary, these mice had calcified lesions in most soft tissues by the age of four months with associated multifactorial morbidity and mortality.

Computed tomography of (A) a male wildtype control mouse at 30 weeks of age, and (B-D) D2,*Ahsg* -/- mice at 20 weeks (B, C) or 35 weeks (D) of age. (A) Mineral in the skeleton, and ingested food-derived mineral speckling in the GI tract. (B) Dorsal view shows calcification of brown adipose tissue in the neck (interscapular) and around the kidneys (arrows). (C) Ventral view shows calcified lesions in brown adipose tissue in the axillae and in the testes (arrows), and in the spleen and pancreas (arrows in D).

## Calcification starts in the microvasculature

We next performed an extensive screening of von Kossa-stained (for mineral) tissue sections derived from 52-week-old DBA/2 *Ahsg*-/- adult mice in order to get further insight into the microstructure and process of lesion formation. From this, we noted conspicuous calcified lesions located in the lumen of the microvasculature. As examples, Fig 2 shows representative photomicrographs of intravascular calcified lesions in the kidney (Fig 2A and 2B), adipose tissue (Fig 2C and 2D) and pancreas (Fig 2E). Red blood cells could be observed immediately adjacent to calcified lesions (Fig 2A–2D), further evidence that they were intravascular. Calcified lesions were observed within arterial vessels including small arterioles (Fig 2B) as well as in small arteries (Fig 2A and 2C). Occasionally, well-rounded globular "stones" were observed (Fig 2E). Haematoxylin/eosin staining showed that all lesions were coagulum/protein-rich (Fig 2F and 2G). In most cases, however, the structure of the lesions was perturbed to some degree by sectioning. Another limitation of routine paraffin histology was that late-stage calcified lesions were accompanied by widespread tissue remodeling and fibrosis, and it was thus difficult to render judgement on the process of primary lesion localization and morphology. To better address this, we performed transmission electron microscopy (TEM) of early-stage calcification in intrascapular brown adipose tissue dissected from 2-week-old fetuin-A-deficient DBA/2 mice, a tissue among the first to be affected by the ectopic calcification (S1A Fig). In agreement with our findings from von Kossa staining of tissue sections from adult mice, we detected calcified, crystalline material within the blood vessels (Fig 3A–3F). Intravascular lesions were detected in microvessels of all sizes ranging from capillaries (diameter 4–20 μm, Fig 3D–3F) to arterioles (diameter 35 μm, Fig 3A–3C). The lesion shown in Fig 3A–3C was surrounded by an endothelium (tunica intima) enclosed by a single layer of smooth muscle cells (tunica media) and the tunica externa, while the intravascular lesions shown in Fig 3D–3F were surrounded by a single layer of endothelium characteristic of capillaries. Occasionally, red blood cells were enclosed by mineral precipitate (Fig 3D and 3E). Intravascular lesions were composed of organic, cellular and electron-dense crystalline structures. To confirm the presence of calcified material in such lesions, we performed selected-area electron diffraction (SAED), X-ray diffraction (XRD) and energy-dispersive X-ray spectroscopy (EDS) on an intravascular lesion, first identified within a microvessel by toluidine blue staining of semi-thin

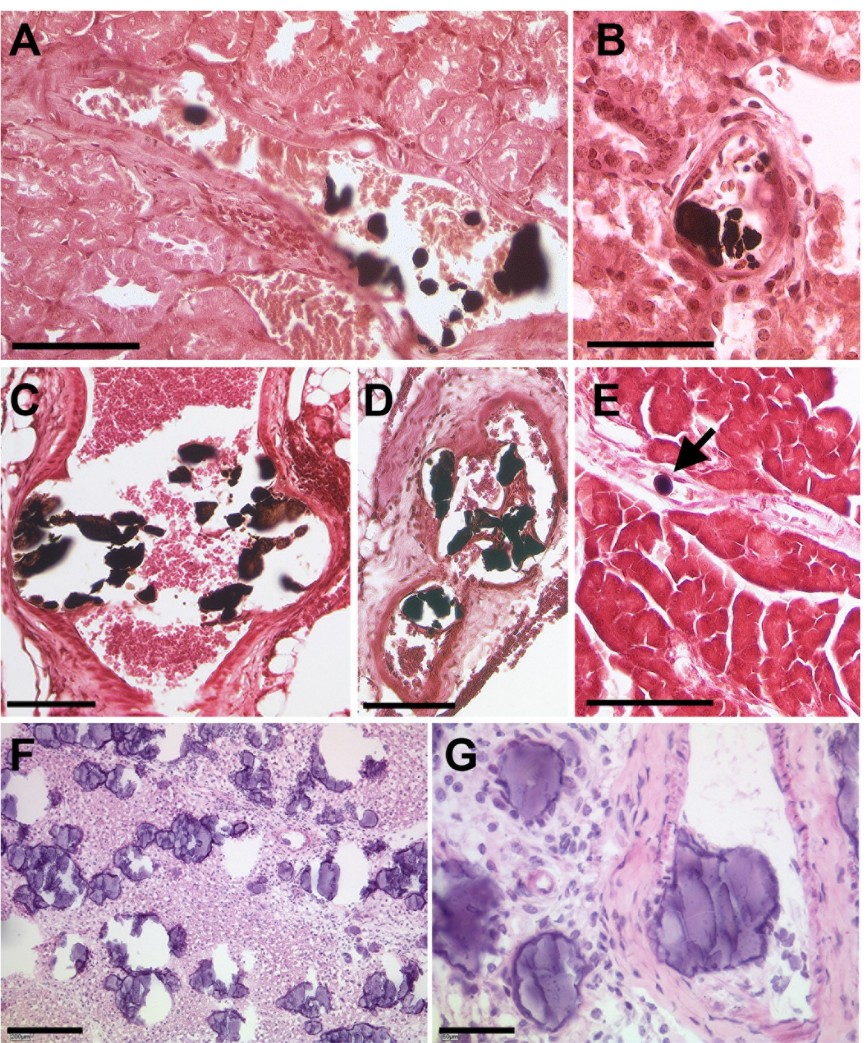

**Fig 2. Calcified lesions within the vasculature.**

sections (Fig 4A). EDS compositional probing of different areas of the lesion found calcium and phosphate as major elements present in the mineral (Fig 4B and 4D). SAED and XRD of the calcified lesion showed diffraction maxima characteristic of a poorly crystallized apatitic mineral phase (Fig 4C, insets) as typically observed in biogenic apatites.

Von Kossa staining for mineral of paraffin sections of kidney (A, B), brown adipose tissue (C, D) and pancreas (E) from a 52-week-old female D2,Ahsg-/- mouse. Mineral deposits stain black (also arrow), where calcified lesions occur frequently within arterioles, partially obstructing the lumen of these vessels. Hematoxylin and Eosin staining of paraffin sections, here of brown adipose tissue (F-G), reveals strong staining of the mineral deposits, indicating also the presence of a protein-rich matrix in these calcified lesions. Scale bars 200 μm (F), 100 μm (A, C-E) and 50 μm (B, G).

Transmission electron micrographs of calcified lesions in brown adipose tissue from a 2-week-old D2,Ahsg-/- mouse. Increasing magnifications (A-C) and additional micrographs (D-F) of lumenal mineral detected within small blood vessels. rbc, red blood cell; bv, blood

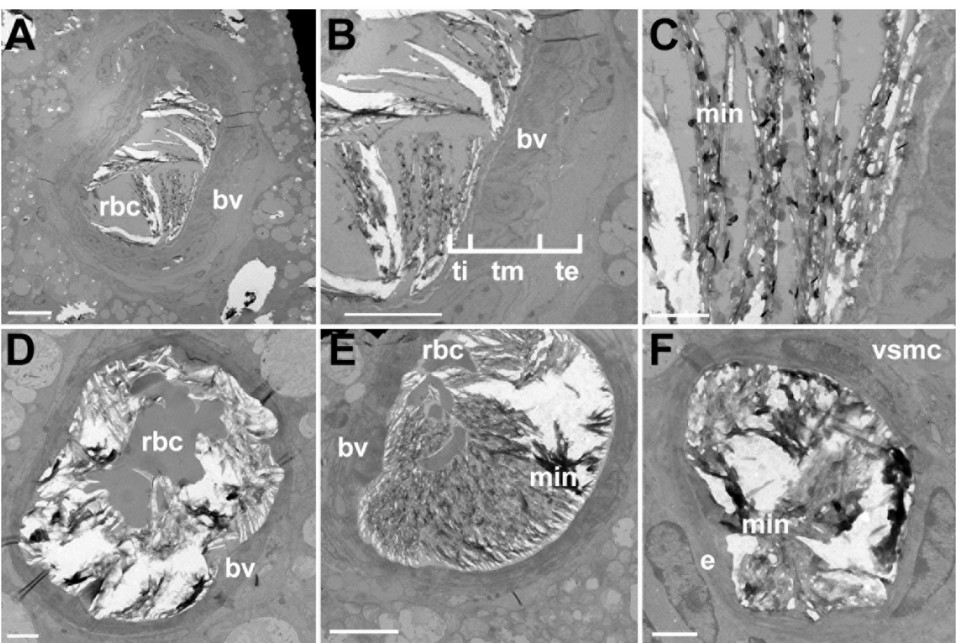

**Fig 3. Electron micrographs of calcified lesions within small vessels.**

vessel wall; ti, tunica intima; tm, tunica media; te, tunica externa; min, mineral; e, endothelium; vsmc, vascular smooth muscle cell. Scale bars: 10 μm (A, B); 2 μm (C, D, F); 5 μm (E).

(A) Calcified lesion within an arteriole identified in a semi-thin plastic section stained with toluidine blue of brown adipose tissue from a 2-week-old D2,*Ahsg*-/- mouse. (B) Inset in (A) from the interior of the arteriole imaged from an adjacent thin section by TEM shows highly contrasted mineral density which were probed for mineral phase characterization. Circled areas show a calcium-phosphate mineral phase of poorly crystalline hydroxyapatite as determined by diffraction maxima obtained by X-ray diffraction and selected-area electron diffraction (C), and by energy-dispersive X-ray spectroscopy (D).

In order to further confirm the localization of the calcified lesions within the microvasculature, we performed immunostaining for endothelial and lymphatic markers, and calcium/mineral-binding proteins, and examined these by fluorescence microscopy. We used two different techniques to detect calcified lesions by immunofluorescence staining. Conventional antibody staining for osteopontin(OPN)–a mineral-binding protein known to be abundant at calcification sites [52, 53]–was used to identify calcified lesions. As a second method, we injected mice with bovine fetuin-A which, because of its strong calcium phosphate-binding properties, accumulates at calcification sites. Immunofluorescence staining for OPN and for injected fetuin-A detected small-diameter and moderate-sized calcified lesions (Fig 5A–5D). This high sensitivity allowed early detection of very small calcified lesions at 2 weeks of age that could not be detected using routine histology or computed tomography. Fig 5A and 5C show particularly well early-stage lesions clearly localized inside capillaries, as shown by surrounding CD31-positive vessels. In contrast, double staining with Lyve-1 revealed that there was no surrounding co-localization with lymphatic vessels (Fig 5D).

Immunofluorescence microscopy of cryosections prepared from lung (A), myocardium (B, D) and brown adipose tissue (C) of D2,*Ahsg*-/- mice. Calcified lesions (arrows) stain positive for endogenous mineral-binding OPN (red in A, B, D) at sites also demarcated by CD31

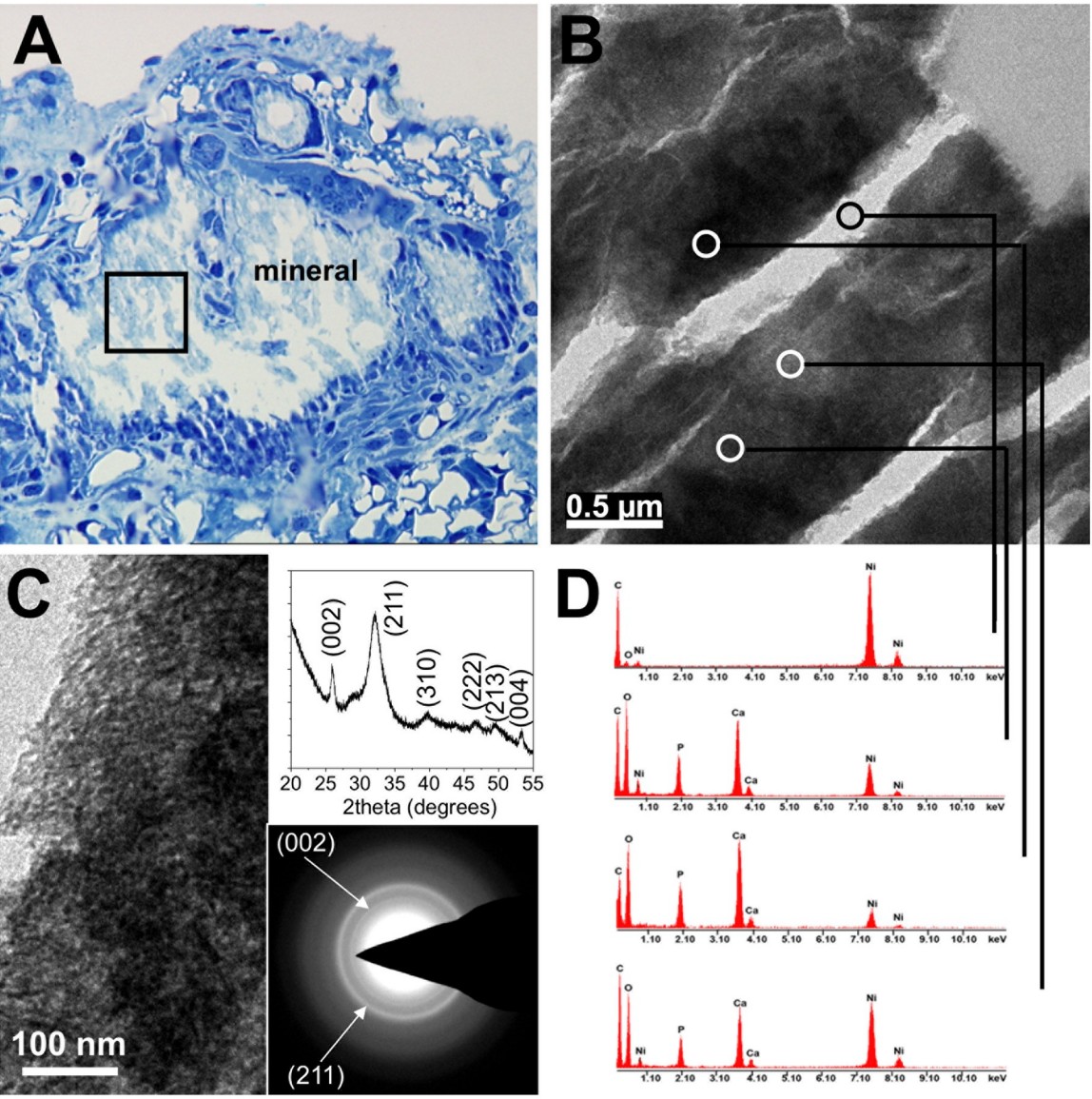

**Fig 4. Mineral analysis of calcified lesions.**

(green) immunoreactivity, an endothelial cell marker (A, B), but not by Lyve-1 (green) a lymph-duct cell marker (D). Injected exogenous fetuin-A binds to mineral in calcified lesions and is readily detected using fetuin-A antibody (C). Even at early states of calcification (2 weeks of age, arrows in A and C), endogenous OPN and injected fetuin-A both demarcate small calcified lesions with high sensitivity (such small lesions are not detected in routine histology and computed tomography) surrounded by CD31-positive endothelial cells. Scale bars equal 50 μm.

## Calcified lesions contain plasma proteins

We next analyzed the composition of dissected calcified lesions in D2,*Ahsg*-/- mice with regard to their protein content. Fig 6A–6C depict millimeter-sized lesions isolated from interscapular

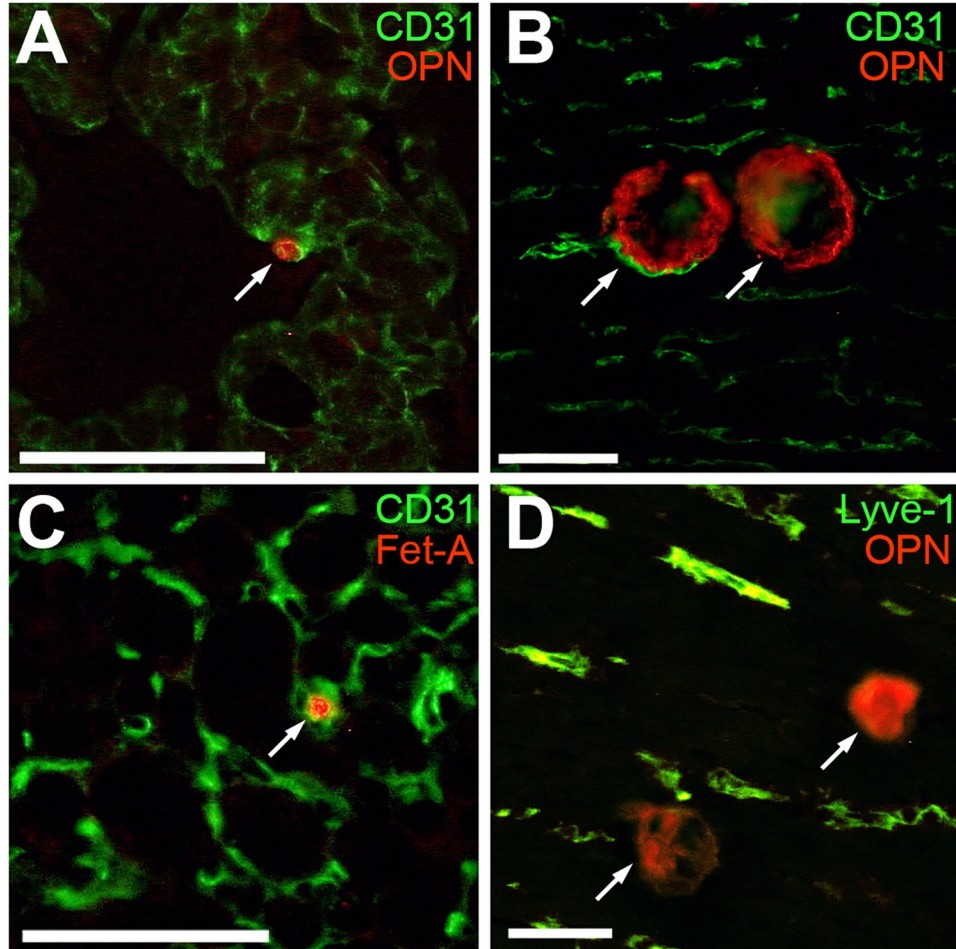

**Fig 5. Co-localization of early-stage calcified lesions and vascular markers.**

brown adipose tissue. Proteins were extracted from such calcified lesions, and intact tissue was also dissected from interscapular brown adipose tissue, heart and skin, and separated using SDS-PAGE (Fig 6D). For mass spectrometry (MS) analysis of the protein content in the calcified lesions, we focused on the most abundant bands of both intact control tissue and extracts of calcified lesions. Association of these proteins with functional pathways revealed that lesions in all organs contained protein components of the coagulation and complement cascade (Tables 1–3). The detection by MS of plasma proteins in calcified lesions was consistent with the histology and TEM results, together indicating that calcified lesions develop within the lumen of microvessels.

Calcified lesions as well as noncalcified adjacent tissue samples from skin and heart were dissected from D2,*Ahsg-/-* mice showing severe ectopic calcification. Proteins were extracted using SDS sample buffer containing EDTA (to dissolve mineral and release bound proteins) and homogenized using a mixer mill. (A-C) Nodular calcified lesions in subcutaneous fat tissue of skin stained with Alizarin Red (A), or they do not stain (B,C). Similar nodules were scraped from interscapular brown adipose tissue (BAT) and dissected from myocardium. (D) SDS-PAGE of samples prepared from serum, intact noncalcified tissue and calcified lesions of skin, heart and brown adipose tissue. M, molecular-weight marker.

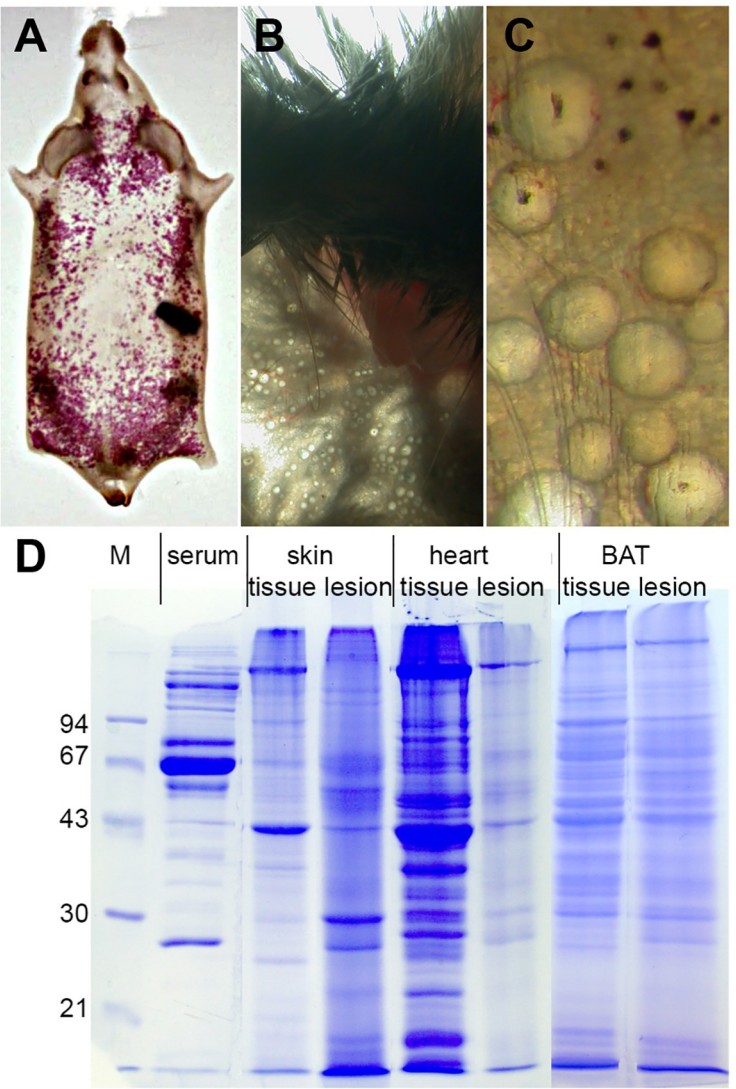

**Fig 6. Samples as prepared for proteomic analysis of calcified lesions.**

## Calcification is not associated with osteogenic differentiation and bone formation

To elucidate the molecular pathways triggering soft tissue calcification in these mice, we performed genome-wide gene expression analysis in DBA/2 wildtype and fetuin-A-deficient mice. Fig 7 shows a volcano plot representation of the gene expression profile of kidney parenchyme and kidney-associated brown adipose tissue dissected from 5–6 week-old mice. In brown adipose tissue, severe calcification is usually observed in 5–6 week-old mice enabling the detection of genes, which are differentially expressed as a result of the calcification (Fig 7B). In the kidney, lesion formation starts considerably later. Thus, differential evaluation of the gene expression pattern in these two tissues may provide insights into the mechanisms preceding the formation of calcified lesions. According to the different calcification states in kidney parenchyme and kidney-associated brown adipose tissue, only minor changes in gene expression were observed in the kidney parenchyme (narrow shape of volcano plot, Fig 7A),

**Table 1. Pathways associated with proteins contained in brown adipose tissue calcified lesions.**

| pathway description | observed proteins | p-value |
|---|---|---|
| Complement and coagulation cascades | 22 | <0.001 |
| Metabolic pathways | 61 | <0.001 |
| Microbial metabolism in diverse environments | 23 | <0.001 |
| Carbon metabolism | 19 | <0.001 |
| Ribosome | 20 | <0.001 |
| Pyruvate metabolism | 13 | <0.001 |
| Staphylococcus aureus infection | 12 | <0.001 |
| Propanoate metabolism | 10 | <0.001 |
| Systemic lupus erythematosus | 14 | <0.001 |
| Citrate cycle (TCA cycle) | 9 | <0.001 |
| Phagosome | 17 | <0.001 |
| Prion diseases | 8 | <0.001 |
| Parkinson s disease | 14 | <0.001 |
| Biosynthesis of amino acids | 10 | <0.001 |
| Valine, leucine and isoleucine degradation | 8 | <0.001 |
| ECM-receptor interaction | 10 | <0.001 |
| Glyoxylate and dicarboxylate metabolism | 6 | <0.001 |
| Protein processing in endoplasmic reticulum | 13 | <0.001 |
| Amoebiasis | 11 | <0.001 |
| Nitrogen metabolism | 5 | <0.001 |
| Huntington s disease | 13 | <0.001 |
| Cysteine and methionine metabolism | 6 | <0.001 |
| Legionellosis | 7 | 0.001 |
| Focal adhesion | 13 | 0.001 |
| Oxidative phosphorylation | 10 | 0.001 |
| PI3K-Akt signaling pathway | 17 | 0.001 |
| Alzheimer s disease | 11 | 0.001 |
| Pentose phosphate pathway | 5 | 0.001 |
| Fatty acid metabolism | 6 | 0.002 |
| Antigen processing and presentation | 7 | 0.002 |
| Pentose and glucuronate interconversions | 5 | 0.003 |
| Fructose and mannose metabolism | 5 | 0.003 |
| Spliceosome | 9 | 0.003 |
| Glycerolipid metabolism | 6 | 0.003 |
| Proximal tubule bicarbonate reclamation | 4 | 0.004 |
| Epstein-Barr virus infection | 11 | 0.005 |
| Pertussis | 6 | 0.010 |
| AMPK signaling pathway | 8 | 0.011 |
| Proteoglycans in cancer | 11 | 0.011 |
| Cardiac muscle contraction | 6 | 0.013 |
| Chagas disease (American trypanosomiasis) | 7 | 0.013 |
| PPAR signaling pathway | 6 | 0.015 |
| Galactose metabolism | 4 | 0.015 |
| Peroxisome | 6 | 0.017 |
| Arginine and proline metabolism | 5 | 0.018 |
| Toxoplasmosis | 7 | 0.019 |
| 2-Oxocarboxylic acid metabolism | 3 | 0.020 |
| Small cell lung cancer | 6 | 0.020 |

*(Continued)*

**Table 1.** (Continued)

| pathway description | observed proteins | p-value |
|---|---|---|
| Non-alcoholic fatty liver disease (NAFLD) | 8 | 0.023 |
| Glycolysis / Gluconeogenesis | 5 | 0.023 |
| Protein digestion and absorption | 6 | 0.024 |
| Tryptophan metabolism | 4 | 0.036 |
| Fatty acid degradation | 4 | 0.039 |
| Fatty acid elongation | 3 | 0.040 |
| Measles | 7 | 0.048 |

**Table 2. Pathways associated with proteins contained in skin calcified lesions.**

| pathway description | observed proteins | p-value |
|---|---|---|
| Phagosome | 31 | <0.001 |
| Complement and coagulation cascades | 14 | <0.001 |
| Staphylococcus aureus infection | 12 | <0.001 |
| Ribosome | 17 | <0.001 |
| Systemic lupus erythematosus | 14 | <0.001 |
| Lysosome | 14 | <0.001 |
| Antigen processing and presentation | 11 | <0.001 |
| Tuberculosis | 16 | <0.001 |
| Amoebiasis | 13 | <0.001 |
| Protein digestion and absorption | 11 | <0.001 |
| Proteasome | 8 | <0.001 |
| Protein processing in endoplasmic reticulum | 13 | <0.001 |
| ECM-receptor interaction | 9 | 0.001 |
| Leukocyte transendothelial migration | 10 | 0.002 |
| Focal adhesion | 13 | 0.003 |
| Rheumatoid arthritis | 8 | 0.003 |
| Regulation of actin cytoskeleton | 13 | 0.004 |
| Spliceosome | 9 | 0.010 |
| PPAR signaling pathway | 7 | 0.010 |
| Proximal tubule bicarbonate reclamation | 4 | 0.010 |
| Legionellosis | 6 | 0.010 |
| Carbon metabolism | 8 | 0.013 |
| Pertussis | 6 | 0.028 |
| Citrate cycle (TCA cycle) | 4 | 0.031 |
| Oxidative phosphorylation | 8 | 0.035 |
| Pancreatic secretion | 7 | 0.035 |
| Nitrogen metabolism | 3 | 0.040 |
| Prion diseases | 4 | 0.042 |
| Hematopoietic cell lineage | 6 | 0.045 |
| Microbial metabolism in diverse environments | 9 | 0.048 |

whereas the severe calcification that occurs in kidney-associated brown adipose tissue caused a dramatic change in gene expression in this tissue, in particular, the upregulation of several genes in D2,*Ahsg-/-* mice (broad shape of volcano plot, Fig 7B). In total, 16 differentially expressed genes (p-value < 0.05) were detected in the kidney (Table 4). In contrast, 395

**Table 3. Pathways associated with proteins contained in heart calcified lesions.**

| pathway description | observed proteins | p-value |
|---|---|---|
| Complement and coagulation cascades | 20 | <0.001 |
| Proteasome | 8 | <0.001 |
| Citrate cycle (TCA cycle) | 7 | <0.001 |
| Prion diseases | 7 | <0.001 |
| Carbon metabolism | 10 | <0.001 |
| Amoebiasis | 10 | <0.001 |
| Hypertrophic cardiomyopathy (HCM) | 8 | <0.001 |
| ECM-receptor interaction | 8 | <0.001 |
| Dilated cardiomyopathy | 8 | <0.001 |
| Metabolic pathways | 26 | <0.001 |
| Staphylococcus aureus infection | 6 | <0.001 |
| Alzheimer s disease | 9 | <0.001 |
| Microbial metabolism in diverse environments | 9 | <0.001 |
| Systemic lupus erythematosus | 7 | <0.001 |
| 2-Oxocarboxylic acid metabolism | 4 | <0.001 |
| PI3K-Akt signaling pathway | 12 | <0.001 |
| Biosynthesis of amino acids | 6 | <0.001 |
| Cardiac muscle contraction | 6 | <0.001 |
| PPAR signaling pathway | 6 | <0.001 |
| Huntington s disease | 8 | 0.001 |
| Parkinson s disease | 7 | 0.001 |
| Focal adhesion | 8 | 0.002 |
| Phagosome | 7 | 0.003 |
| Arrhythmogenic right ventricular cardiomyopathy (ARVC) | 5 | 0.003 |
| Fatty acid degradation | 4 | 0.004 |
| Peroxisome | 5 | 0.004 |
| Oxidative phosphorylation | 6 | 0.005 |
| Fatty acid metabolism | 4 | 0.006 |
| Epstein-Barr virus infection | 7 | 0.008 |
| Proteoglycans in cancer | 7 | 0.014 |
| Antigen processing and presentation | 4 | 0.021 |
| Viral carcinogenesis | 6 | 0.035 |
| Small cell lung cancer | 4 | 0.035 |
| Adrenergic signaling in cardiomyocytes | 5 | 0.038 |
| Protein digestion and absorption | 4 | 0.039 |

differentially expressed genes were found in brown adipose tissue. The significantly regulated genes with a fold change above four (log-ratio >3) are summarized in Table 5.

Volcano plot representation of gene expression in kidney (A) and brown fat (B) dissected from the region of the kidney pelvis. Differential gene expression between fetuin-A-deficient and wildtype mice is shown as log-ratio on the x-axis; negative values represent higher expression in D2,*Ahsg*-/- mice, and positive values represent higher expression in wildtype mice. The y-axis encodes the probability for differential regulation calculated by Bayesian statistics in the Limma package under Bioconductor. Each dot denotes a probe set, and probe sets with the highest probability score are depicted in blue, probe sets with highest log-ratio are depicted in red, and marked probe sets are labeled with the appropriate gene name. Slc15a2 = solute carrier family 15 (H+/peptide transporter), member 2; Pdxdc1 = pyridoxal-dependent

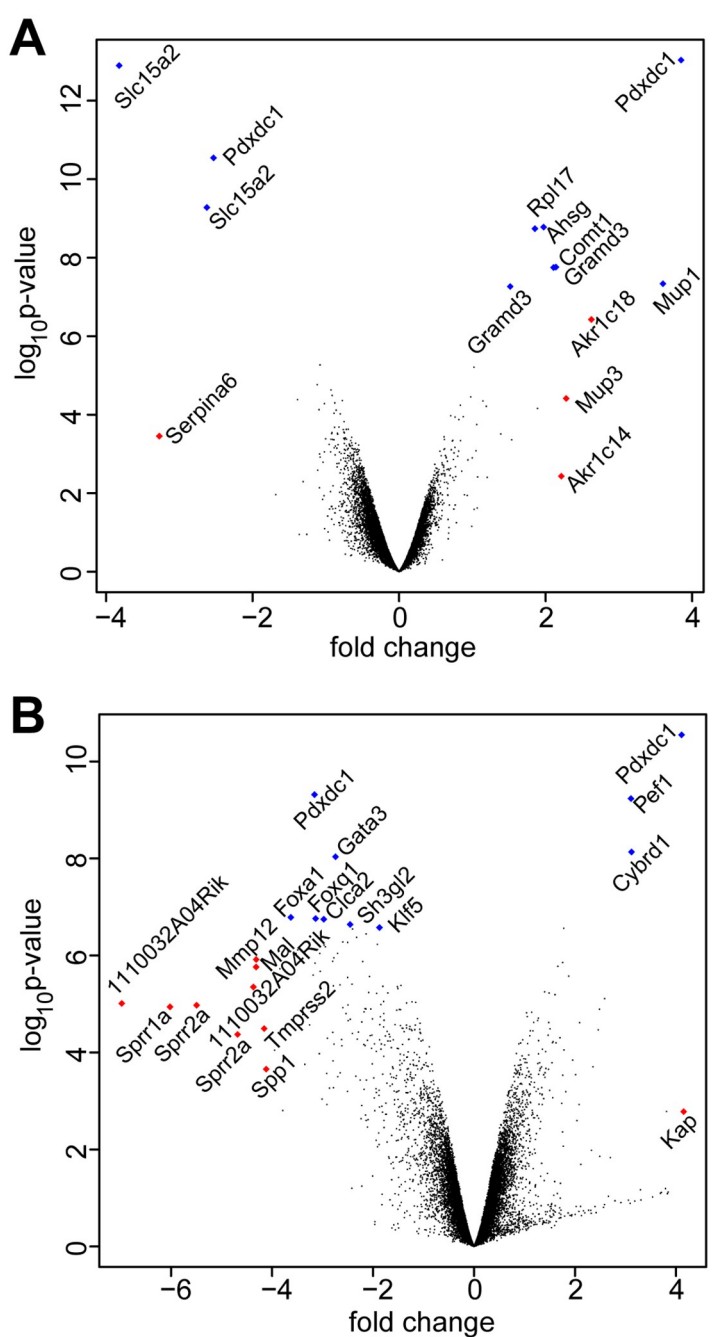

**Fig 7. Gene expression in 6-week-old DBA/2 wildtype and Ahsg-/- mice.**

decarboxylase domain containing 1; Serpina6 = serine (or cysteine) peptidase inhibitor, clade A, member 6; Rpl17 = ribosomal protein L17; Ahsg = alpha-2-HS-glycoprotein; Comt1 = catechol-O-methyltransferase 1; Gramd3 = GRAM domain containing 3; Akr1c18 = aldo-keto reductase family 1, member C18; Mup1 = major urinary protein 1; Mup3 = major urinary protein 3; Akr1c14 = aldo-keto reductase family 1, member C14; Gata3 = GATA binding protein 3; Foxa1 = forkhead box A1; Foxq1 = forkhead box Q1; Clca2 = chloride channel accessory 2; Sh3gl2 = SH3-domain GRB2-like 2; Klf5 = Kruppel-like

**Table 4. Differential expression in kidneys of 5-week-old D2, *Ahsg*-/- mice.**

| probe set | symbol | p-value | log-ratio |
|---|---|---|---|
| induced genes | | | |
| solute carrier family 15 (H+/peptide transporter), member 2 | Slc15a2 | < 0.001 | -3.821 |
| solute carrier family 15 (H+/peptide transporter), member 2 | Slc15a2 | < 0.001 | -2.624 |
| pyridoxal-dependent decarboxylase domain containing 1 | Pdxdc1 | < 0.001 | -2.532 |
| angiotensin I converting enzyme (peptidyl-dipeptidase A) 2 | Ace2 | 0.030 | -1.117 |
| ubiquitin specific peptidase 1 | Usp1 | 0.011 | -1.079 |
| pyridoxal-dependent decarboxylase domain containing 1 | Pdxdc1 | 0.038 | -0.929 |
| sulfotransferase family 3A, member 1 | Sult3a1 | 0.049 | -0.913 |
| repressed genes | | | |
| pyridoxal-dependent decarboxylase domain containing 1 | Pdxdc1 | < 0.001 | 3.849 |
| major urinary protein 1 | Mup1 | < 0.001 | 3.601 |
| aldo-keto reductase family 1, member C18 | Akr1c18 | 0.001 | 2.625 |
| catechol-O-methyltransferase 1 | Comt1 | < 0.001 | 2.141 |
| GRAM domain containing 3 | Gramd3 | < 0.001 | 2.109 |
| alpha-2-HS-glycoprotein | Ahsg | < 0.001 | 1.973 |
| ribosomal protein L17 | Rpl17 | < 0.001 | 1.854 |
| GRAM domain containing 3 | Gramd3 | < 0.001 | 1.517 |
| ring finger protein 41 | Rnf41 | 0.012 | 1.022 |

The table shows probe sets, which were significantly (p-value < 0.05) differentially expressed in kidneys dissected from 5-week-old DBA/2 wildtype and fetuin-A deficient mice. Bayesian statistics was used for calculation of probabilities (p-value) and log-ratio, negative log-ratio encode gene induction of the particular probe-set in fetuin-A deficient mice, positive values denote gene repression. Note that gene names are given for each probe set, double entries may occur in case of genes which are represented by several probe sets encoding different regions or splice variants of the gene.

factor 5 (intestinal); Mmp12 = matrix metallopeptidase 12; Mal = myelin and lymphocyte protein, T-cell differentiation protein; 1110032A04Rik = RIKEN cDNA 1110032A04 gene; Sprr1a = small proline-rich protein 1A; Sprr2a = small proline-rich protein 2A; Tmprss2 = transmembrane protease, serine 2; Spp1 = secreted phosphoprotein 1; Pef1 = penta-EF hand domain containing 1; Cybrd1 = cytochrome b reductase 1; Kap = kidney androgen regulated protein.

The two most highly upregulated probe sets in the kidney encoded for solute carrier family 15 (H+/peptide transporter), member 2 (Slc15a2) and were 14.1 and 6.2-fold increased in *Ahsg*-/- kidneys, respectively. Slc15a2 –also known as peptide transporter 2 (PEPT2)–is expressed predominantly in the kidney and has been shown to be a proton-dependent transporter of di- and tri-peptides [54]. Remarkably, neither this transporter nor any other of the differentially regulated genes, was related to mineral transport or mineralization processes, suggesting that calcification is not driven by genetic reprogramming of cells towards an osteoblastic phenotype.

In brown adipose tissue, 27 probe sets were more than 6-fold increased in D2,*Ahsg*-/- mice. Among the ten most highly upregulated probe sets, two members of the small proline rich (SPRR) protein family were found, a protein family previously associated with various inflammatory diseases, cellular stress and repair [55]. *Sprr1a* was 64.9-fold increased, *Sprr2a* was represented by two probe sets with 45.1- and 25.7-fold upregulation in *Ahsg*-/- mice, respectively. The matrix metallopeptidase 12 (*Mmp12*), involved in extracellular remodeling, showed a fold change of 19.9. Secreted phosphoprotein 1 (*Spp1*, OPN), with a fold change of 17.3, was detected as another highly upregulated probe set. *Spp1* is a secreted multifunctional glyco-phosphoprotein involved in mineral metabolism. *Spp1* expression levels are found to be

**Table 5. Differential expression in adipose tissue of 6-week-old D2, *Ahsg*-/- mice.**

| probe set | symbol | p-value | log-ratio |
|---|---|---|---|
| induced genes | | | |
| RIKEN cDNA 1110032A04 gene | 1110032A04Rik | 0.004 | -6.977 |
| small proline-rich protein 1A | Sprr1a | 0.005 | -6.021 |
| small proline-rich protein 2A | Sprr2a | 0.005 | -5.495 |
| small proline-rich protein 2A | Sprr2a | 0.009 | -4.683 |
| RIKEN cDNA 1110032A04 gene | 1110032A04Rik | 0.003 | -4.369 |
| myelin and lymphocyte protein, T-cell differentiation protein | Mal | 0.001 | -4.316 |
| matrix metallopeptidase 12 | Mmp12 | 0.001 | -4.315 |
| transmembrane protease, serine 2 | Tmprss2 | 0.008 | -4.157 |
| secreted phosphoprotein 1 | Spp1 | 0.024 | -4.116 |
| uroplakin 3A | Upk3a | 0.014 | -3.949 |
| lymphocyte antigen 6 complex, locus D | Ly6d | 0.002 | -3.845 |
| sorting nexin 31 | Snx31 | 0.008 | -3.728 |
| forkhead box A1 | Foxa1 | 0.001 | -3.628 |
| myelin protein zero | Mpz | 0.001 | -3.525 |
| Purkinje cell protein 4 | Pcp4 | 0.001 | -3.493 |
| prostate stem cell antigen | Psca | 0.008 | -3.436 |
| carnitine palmitoyltransferase 1b, muscle | Cpt1b | 0.006 | -3.390 |
| WAP four-disulfide core domain 2 | Wfdc2 | 0.022 | -3.351 |
| cadherin 1 | Cdh1 | 0.007 | -3.256 |
| claudin 4 | Cldn4 | 0.002 | -3.185 |
| pyridoxal-dependent decarboxylase domain containing 1 | Pdxdc1 | < 0.001 | -3.159 |
| myelin basic protein | Mbp | 0.001 | -3.148 |
| forkhead box Q1 | Foxq1 | 0.001 | -3.137 |
| involucrin | Ivl | 0.005 | -3.078 |
| transmembrane protease, serine 2 | Tmprss2 | 0.016 | -3.063 |
| FXYD domain-containing ion transport regulator 3 | Fxyd3 | 0.001 | -3.056 |
| keratin 18 | Krt18 | 0.002 | -3.046 |
| repressed genes | | | |
| pyridoxal-dependent decarboxylase domain containing 1 | Pdxdc1 | < 0.001 | 4.111 |
| cytochrome b reductase 1 | Cybrd1 | < 0.001 | 3.119 |
| penta-EF hand domain containing 1 | Pef1 | < 0.001 | 3.108 |

The table shows probe sets, which were significantly (p-value < 0.05) differentially expressed in adipose tissue. Bayesian statistics was used for calculation of probabilities (p-value) and log-ratio, negative log-ratio encode gene induction of the particular probe-set in fetuin-A deficient mice, positive values denote gene repression. The table shows most highly differentially regulated genes with a log-ratio above 3. Note that gene names are given for each probe set, double entries may occur in case of genes which are represented by several probe sets encoding different regions or splice variants of the gene.

elevated in several chronic inflammatory disease pathologies [56] indicating chronic inflammation also caused by calcification. All significantly regulated probe sets were screened for biological function using the KEGG gene set. Table 6 lists ten gene sets, which were significantly overrepresented among the tested probe sets. Several pathways related to tissue remodeling, ECM-receptor interaction, cell cycle, p53 signaling, cell adhesion molecules and the Notch signaling pathway were amongst the differentially regulated pathways. Notably, no probe set directly associated with osteogenic differentiation or bone formation was differentially regulated.

**Table 6. Pathway analysis of significant regulated genes in brown adipose tissue of D2, *Ahsg*-/- mice.**

| KEGG pathway | size | p-value |
|---|---|---|
| Focal adhesion | 174 | 0.001 |
| ECM-receptor interaction | 74 | 0.004 |
| Cell cycle | 100 | 0.005 |
| p53 signaling pathway | 54 | 0.005 |
| Small cell lung cancer | 79 | 0.006 |
| Selenoamino acid metabolism | 25 | 0.015 |
| Aminophosphonate metabolism | 13 | 0.03 |
| Cell adhesion molecules (CAMs) | 115 | 0.034 |
| Notch signaling pathway | 37 | 0.043 |
| Bladder cancer | 37 | 0.043 |

Differentially expressed probe sets in adipose tissue were tested of over-representation of KEGG gene sets. For each gene set the name of the pathway and the corresponding size is given. The probability of overrepresentation of the gene set is assumed by the p-value.

## Consequences of intralumenal calcification; reduced growth and premature death in fetuin-A-deficient DBA/2 mice

Over the course of several years we recorded the weight of a total of >750 random selected animals that were analyzed at ages 7–450 days in this or in related studies. Fig 8A and 8B show that bodyweights of C57BL/6 (B6) and DBA/2 (D2) wildtype mice in our colony plateaued at 32 and 33 grams, respectively. Fetuin-A-deficient B6 littermate mice similarly attained bodyweights of 34 grams. In contrast, fetuin-A-deficient DBA/2 mice diverged in bodyweight from about 3 weeks onward, and attained reduced bodyweights of 26±3 grams at the end of the observation period (Fig 8A). Importantly, we observed along the timeline outliers with less than half normal body weight, which had severe ectopic calcification as shown in S1A–S1H Fig. Fig 8C and 8D show the 12-month survival rates in B6,wt, B6,ko and D2,wt mice were higher than 90% regardless of sex. In contrast, the survival rates in D2,ko mice were reduced at 86% in male and 61% in female mice.

A, B) Randomly selected mice from the colony were weighed each at the day they were sacrificed for imaging or tissue harvest. Note that overall, B6,wt (n = 213) B6,ko (n = 205), and D2,wt mice (n = 138) all attained a maximum weight of 30±5 gram while D2,ko mice (n = 244) weighed in at roughly five gram lighter. The 95% confidence intervals of body weights are plotted as grey lines flanking the black fitted weight curves in each case. Outlier D2,ko mice with less than half normal body weight invariably were the most calcified upon autopsy (Fig 6 and S1 Fig). C, D, The 12-month survival rate in B6,wt, B6,ko and D2,wt mice was higher than 90% regardless of sex and genetic background. In contrast the survival rates in D2,ko mice were reduced at 86% in male and 61% in female mice.

We analyzed by echocardiography the cardiac output of isoflurane anesthetized mice at 4, 8, and 12 months age (S1J Fig). Wildtype mice increased their cardiac output from 7.5±2.2 ml/min at 4 months age to 12.5±5 ml/min at 12 months in good agreement with published values for DBA/2 mice. In contrast, fetuin-A-deficient mice had progressively reduced cardiac output ranging from 7±2 ml/min at four months to 6±1 ml/min at 8 months, and 5±1 ml/min at 12 months, respectively, suggesting worsening of cardiac function with age. Thus D2,ko mice most likely died of cardiac failure as heart function progressively deteriorated with calcification as published [32].

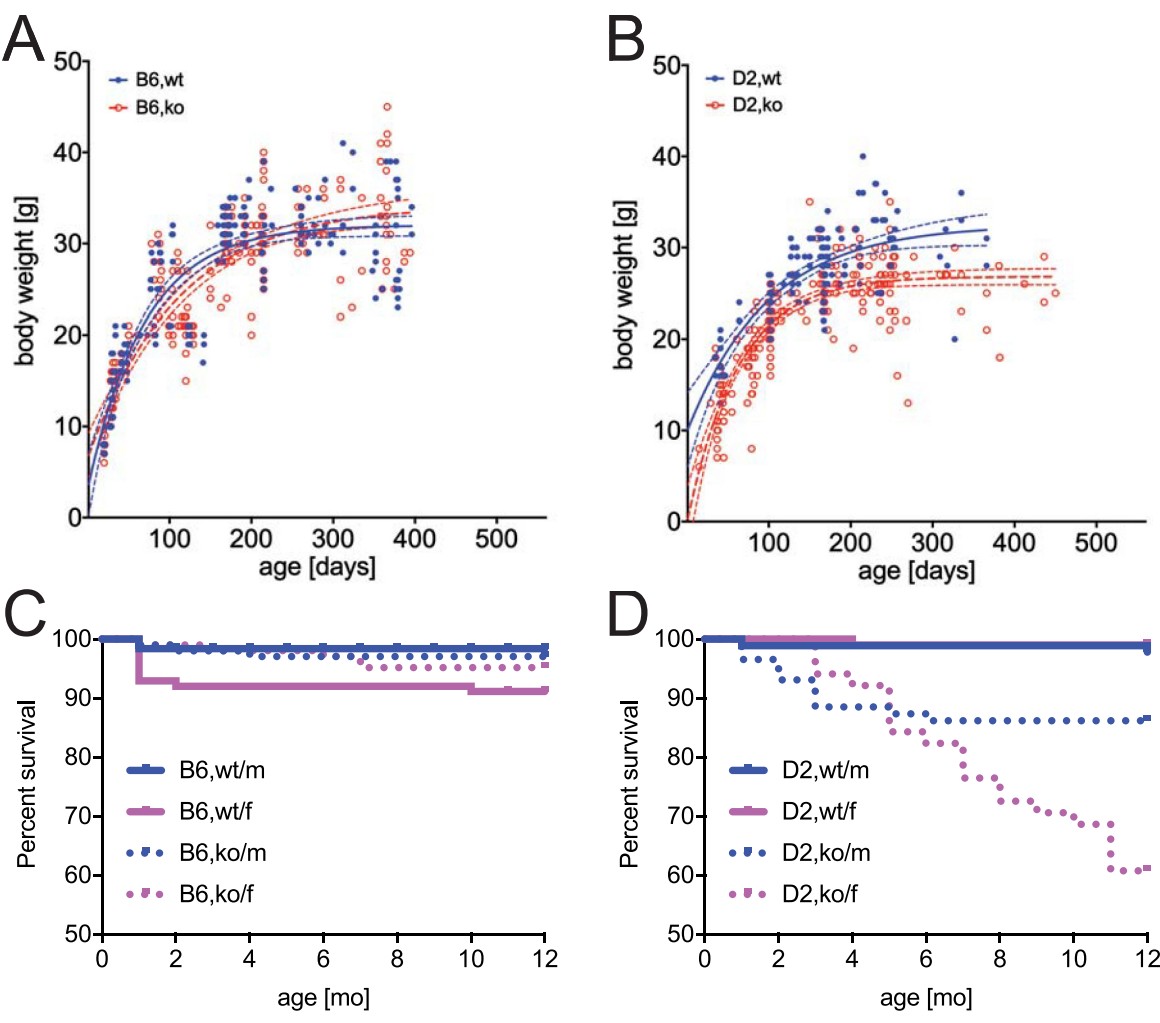

**Fig 8. Impaired growth and premature death in fetuin-A deficient DBA/2 mice.**

## Discussion

In mice, the combination of the calcification-prone genetic background DBA/2 and fetuin-A deficiency leads to a spontaneous, severe ectopic calcification, which occurs on a systemic level [29]. The severe calcification phenotype in mice causes premature death attributable to secondary failure of several organs including cardiac dysfunction, as shown in previous work from our group [32]. We regard the sexual dimorphism shown in Fig 8B and 8D as a consequence of sex-specific organ failure secondary to calcification. This is also supported by our gene expression analysis, where sex specific differences where considered in the analysis, yet they did not reveal any differences. There are several reasons why females may react different than males. The survival curves shown in Fig 8D contain homozygous female breeders, which generally stop breeding after one or two pregnancies and show extensive calcification of their reproductive organs. We hypothesize that pregnancy and lactation, with the ensuing mobilization and transport of mineral to meet the needs of fetuses and pups, put extra stress on the extracellular mineral ion-buffering system of fetuin A-mediated CPP formation and that this may contribute to the higher female lethality. In fact, we reported in the original description of *Ahsg*-knockout mice on a mixed C57BL/6 X Sv129 background that spontaneous ectopic

calcification appeared only in female ex-breeders [57]. The full penetrance and severe expression of the calcification phenotype renders D2,*Ahsg*-/- mice ideal for studying mechanisms and therapeutic approaches for calcification diseases. A concurrent study from our laboratory has shown that the severe soft tissue calcification in the mice is governed by fetuin-A, pyrophosphate and magnesium, and can in fact be prevented by prophylactic supplementation of each one of these major systemic inhibitors of calcification [30]. The aim of this present study was to analyze the early events leading to soft tissue calcification in these mice. We first used computed tomography to survey and identify organs affected by ectopic calcification in living mice. In this way, we detected soft tissue calcification in heart, lung, spleen, pancreas, skin, and gonads, and for the first time, in brown adipose tissue (Fig 1). The high number of affected organs underscores the systemic nature of the calcification phenotype. We focused on early-stage lesions detected by von Kossa staining and histology (Fig 2), TEM (Fig 3), and immuno-fluorescence microscopy (Fig 5). Notably, affected tissues are all known to be highly vascularized with capillaries (capillarized). This particular observation is significant, because we show here that calcified lesions start in the lumen of microvessels.

We further show by gene expression analysis of calcified brown adipose tissue that the formation of calcified lesions triggers robust tissue remodeling. Thus, the morphology of the tissue in proximity to the calcified lesions is quickly and thoroughly remodeled, precluding a detailed analysis of calcification-triggering events. To overcome this limitation, monitoring of early-stage lesions or even the growth process of small calcified lesions would be preferable, but this is not yet accessible because of the limited resolution of *in vivo* imaging methods and devices for small rodents, and because of the fast spreading of calcified lesions once started. Despite these limitations in methodology, we did in fact observe many intravascular lesions using advanced imaging methods on the harvested tissues, including electron microscopy, supporting the notion that calcification in D2,*Ahsg*-/- mice starts in the lumen of the vasculature. This contrasts to some extent with currently prevailing dogma (despite reservations on this, ref. [58]) stating that ectopic calcification involves osteo/chondral differentiation of resident cells, therefore recapitulating bone formation. Instead, calcification in D2,*Ahsg*-/- mice emphasizes the regulation by fetuin-A of mineralization deriving from complex solutions, a notion that has been repeatedly demonstrated in collagen mineralization [59, 60]. Recent work on crystal growth challenges classical nucleation theory and stresses the importance of particle attachment in synthetic, biogenic, and geologic environments [61]. Synthetic and natural polymers have been shown by many laboratories to maintain high supersaturation by forming colloidal particles, and to control mineralization. Mineral-containing particles have recently also been described to provide bulk mineral for bone mineralization [62, 63] and cardiovascular calcification [64].

Fetuin-A has been shown to stabilize vesicles mediating smooth muscle cell calcification particles [65], a process reminiscent of extracellular vesicle-derived microcalcification in atherosclerotic plaques [66]. We suggest here that fetuin-A is a prototype natural polymer involved in colloidal mineral particle formation and stabilization in vertebrates. Fetuin-A is an abundant plasma protein, and low plasma levels have been variably related to calcification-associated diseases [19–24]. We and others have shown that fetuin-A is a main component of colloidal mineral-protein particles found in supersaturated calcium-phosphate solution *in vitro* [6, 7] and in serum of CKD patients [11, 67]. Most importantly, fetuin-A is crucial for the formation and stabilization of calciprotein particles (CPP) [4, 6], which subsequently mediate the clearance of excess calcium-phosphate by macrophages in liver and spleen [3, 5]. Considering the high serum calcium and phosphate concentration, even under physiological conditions, it is not surprising that fetuin-A deficiency results in precipitation of calcium-phosphate directly from the circulation. Precipitating calcium-phosphate

mineral may then serve as a solid phase for blood coagulation explaining the presence of coagulation factors detected by protein MS in calcified lesions. Furthermore, our analysis of the protein content of calcified lesions revealed a striking similarity to serum granulations derived from human or bovine serum mixed with calcium, phosphate, or both, which were studied by Young and colleagues [7, 68]. The granules contained various plasma proteins, including components of the complement system, all likewise detected in calcified lesions of D2,*Ahsg-/-* mice (Fig 5, Tables 1–3, S1 Table). The striking similarity in protein content of serum granules and calcified lesions studied here suggests similar mechanisms of mineral formation and most importantly the participation of serum. At this time, it is not clear whether the protein composition in calcified lesions merely reflects the relative abundance of proteins in the given environment as previously discussed by Young et al. [7], or if there is a selective enrichment of the identified proteins.

Ectopic calcification has been studied in great detail in association with atherosclerosis and vascular calcification. The identification of matrix vesicles in calcified vasculature [69–71], the detection of bone-associated proteins in human atherosclerotic plaques [72, 73], and the overexpression of several mineralization-associated transcription factors [74–77] led to the view that calcification in general comprises an active biological process including osteogenic transdifferentiation of vascular smooth muscle cells. A similar mechanism was suggested for valvular calcification [78]. In contrast, a DNA microarray analysis of human calcified aortic valves detected elevated inflammatory markers but no upregulation of osteoblastic markers [79]. In addition, it was shown that the deposition of calcium-phosphate was passive and the osteogenic differentiation of VSMCs happened after the initial calcification [58, 80]. These findings were confirmed in breast arterial calcification in CKD patients and in arterial calcification of 5/6[th] nephrectomized rats on high phosphate diet [81–83]. Thus, it seems that osteochondral cell differentiation may also be a consequence, rather than a cause, of calcification in soft tissues. With this in mind, we interrogated osteogenic and other gene regulatory networks by a genome-wide gene expression study in kidney and brown adipose tissue dissected from DBA/2 wildtype and *Ahsg-/-* mice. The kidney, when examined in a "pre-calcification state," essentially shows no differentially expressed genes suggesting a process of chemical mineral precipitation rather than bone formation processes involving different differentiated cell states. In late-stage ectopic calcification, as was analyzed in brown adipose tissue, we likewise failed to detect any genes or pathways related to mineralization or osteogenic processes. Instead, we identified 395 differentially regulated genes, most of which were upregulated in the D2,*Ahsg-/-* mouse, and which were associated with inflammatory signaling and tissue remodeling processes.

Among the most highly upregulated genes we identified two members of the small proline-rich protein family, SPRR1A and SPRR2A. Both have been associated with various inflammatory diseases including inflammatory skin diseases [84], allergic inflammation [85] and fibrosis [55]. In addition, a stress-related induction of *Sprr1a* in cardiomyocytes and a protective effect against ischemic injury was described [86]. Thus, increased expression of *Sprr1a* in calcified brown adipose tissue in *Ahsg-/-* mice may reflect the pro-fibrotic stimulus of calcification. A 17.3-fold increase in *Spp1*/OPN expression was detected confirming the previous detection of OPN in calcified lesions of D2,*Ahsg-/-* mice [53], in uremic C57BL/6, *Ahsg-/-* mice [87], and in calcified human coronary arteries [72, 88]. It was shown that calcification-associated OPN is secreted by macrophages, which commonly accumulates in calcified lesions [52, 53]. Various functions of OPN have been proposed. Steitz and colleagues reported that OPN physiologically blocked crystal growth by induction of carbonic anhydrase 2 and thus by acidification of the extracellular milieu [89]. An inhibitory role for OPN was confirmed in MGP-deficient mice where genetic ablation of OPN further enhanced the

calcification phenotype [90]. OPN accumulation was however not associated with reduced levels of calcification in D2,*Ahsg*-/- mice. Thus, OPN expression seemed to be reactive rather than preventive. Beside its role in mineralization, OPN has been described in association with a broad spectrum of immunomodulatory functions including chemotactic macrophage infiltration [56]. Thus, OPN expression in calcified brown adipose tissue may reflect a role in the recruitment of macrophages and leukocytes. Another highly upregulated gene in calcified brown adipose tissue was MMP12, which cleaves CXC type chemokines and may therefore play a key role in regulation of active inflammatory responses by terminating the influx of polymorphonuclear leukocytes [91]. Here, an increased MMP12 expression may explain the relatively moderate degree of calcification-associated inflammation (S2 Fig); while macrophage accumulation was observed in most the cases, polymorphonuclear leukocytes were only detected occasionally at calcification sites.

In conclusion, our gene expression analysis revealed an activation of genes involved in tissue remodeling and inflammation in late-stage calcification. The microarray analysis presented here did not indicate that calcification in D2,*Ahsg*-/- mice involves any osteogenic conversion of cells. Instead, our data strongly supports the notion that fetuin-A deficiency in D2,*Ahsg*-/- mice triggers systemic ectopic calcification in the microvasculature. Thus, chemical mineral precipitation in blood causes, blood clotting, ischemia, tissue necrosis and fibrotic-calcific remodeling in fetuin-A-deficient DBA/2 mice.

## Supporting information

**S1 Fig. Nodular calcified lesions and reduced cardiac output in fetuin-A deficient DBA/2 mice.** In unstained tissues, the nodular calcified lesions appear off-whitish, semi-transparent granules of sub-millimeter size. Nodules can be harvested mechanically by scraping of (A) brown adipose tissue in the neck, or from the subcutaneous fat layer of the skin (see main-text Fig 6A–6C). Nodules are present in the kidney fat pads (B), but not the kidney pelvis, of 6-week-old a D2,*Ahsg*-/- mice, ventricular wall and the atrium (C), lung tissue (D), spleen (E), pancreas (F), 16-week old kidney pelvis (G), and ovaries (H). J, Echocardiography showed that cardiac output was reduced in fetuin-A-deficient DBA/2 mice compared to wildtype mice at all ages measured.
(JPG)

**S2 Fig. Early- and late-stage calcification.** Representative von Kossa staining for mineral in paraffin sections of kidney (A) and brown adipose tissue (B) from a 6-week-old D2,*Ahsg*-/- mouse. Calcified lesions, indicated by black staining (arrows) can be observed in brown adipose tissue but not in the kidney parenchyma, reflecting the different calcification state of both tissues. Scale bars 200 μm.
(JPG)

**S3 Fig. Innate immunity marker expression within calcified lesions.** Paraffin sections of 11-week-old D2,*Ahsg*-/- mice were prepared from myocardium (A-F) or lung (G-I) tissue. Sections were stained with antibodies against the macrophage surface marker F4/80 (A, B, G), neutrophil myeloperoxidase (C, D, H) and the monocyte/granulocyte marker Ly-6G and Ly-6C (Gr-1). Scale bars 50 μm.
(JPG)

**S1 Movie. Post-mortem computed tomography showing ectopic calcification in the heart of a D2,*Ahsg*-/- mouse at 16 weeks of age.**
(AVI)

**S2 Movie. Post-mortem computed tomography showing ectopic calcification in the lung from a D2,*Ahsg*-/- mouse at 16 weeks of age.**
(AVI)

**S3 Movie. Post-mortem computed tomography of a kidney dissected from a D2,*Ahsg* -/- mice at age 16-weeks.** Spherical calcified lesions are detected in the kidney cortex. Note that adipose tissue localized in the kidney pelvis region is completely replaced by electron dense calcified lesions.
(AVI)

**S4 Movie. Post-mortem computed tomography showing ectopic calcification of testis from a D2,*Ahsg*-/- mouse at 16 weeks of age.**
(AVI)

**S5 Movie. Post-mortem computed tomography showing ectopic calcification of skin from a D2,*Ahsg*-/- mouse at 16 weeks of age.**
(AVI)

**S6 Movie. Post-mortem computed tomography showing ectopic calcification of the spleen from a D2,*Ahsg*-/- mouse at 16 weeks of age.**
(AVI)

**S7 Movie. Post-mortem computed tomography showing ectopic calcification of pancreas from a D2,*Ahsg*-/- mouse at 16 weeks of age.**
(AVI)

**S1 Table. Underlying data for Fig 8 and S1J Fig.**
(XLSX)

## Acknowledgments

We thank the Genomics-facility of IZKF Aachen for performing the gene expression arrays.

## Author Contributions

**Data curation:** Ulrike Kusebauch, Robert L. Moritz.

**Funding acquisition:** Willi Jahnen-Dechent.

**Investigation:** Marietta Herrmann, Anne Babler, Irina Moshkova, Felix Gremse, Fabian Kiessling, Ulrike Kusebauch, Valentin Nelea, Robert L. Moritz, Marc D. McKee.

**Methodology:** Felix Gremse, Fabian Kiessling.

**Supervision:** Willi Jahnen-Dechent.

**Validation:** Willi Jahnen-Dechent.

**Writing – original draft:** Marietta Herrmann, Willi Jahnen-Dechent.

**Writing – review & editing:** Marietta Herrmann, Rafael Kramann, Marc D. McKee, Willi Jahnen-Dechent.

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
