## [Decision Letter · Decision Letter 0]

28 Oct 2019

PONE-D-19-27423

Microvasculopathy, Luminal Calcification and Premature Aging in Fetuin-A Deficient Mice

PLOS ONE

Dear Prof Jahnen-Dechent,

Thank you for submitting your manuscript to PLOS ONE. After careful consideration, we feel that it has merit but does not fully meet PLOS ONE’s publication criteria as it currently stands. Therefore, we invite you to submit a revised version of the manuscript that addresses the points raised during the review process.

We would appreciate receiving your revised manuscript by Dec 12 2019 11:59PM. To enhance the reproducibility of your results, we recommend that if applicable you deposit your laboratory protocols in protocols.io, where a protocol can be assigned its own identifier (DOI) such that it can be cited independently in the future. For instructions see: http://journals.plos.org/plosone/s/submission-guidelines#loc-laboratory-protocols

We look forward to receiving your revised manuscript.

Kind regards,

Elena Aikawa, MD PhD

Academic Editor

PLOS ONE

Journal Requirements:

2. Thank you for your submission to PLOS ONE. We note that your study design included a survival study and we would therefore ask that you please report additional details in your Methods section regarding animal care and use for the survival study, as per our editorial guidelines (http://journals.plos.org/plosone/s/submission-guidelines#loc-humane-endpoints).      

For easy reference, we have attached a checklist that may be relevant for your submission. Please complete all items on the checklist at the following link:   http://journals.plos.org/plosone/s/file?id=bb1d/plos-one-humane-endpoints-checklist.docx         

Please upload the completed checklist as file type “Other” when resubmitting your manuscript. This document is for internal journal use only and will not be published if your article is accepted. We very much appreciate your attention to these requests and support of improved reporting standards in PLOS ONE submissions.

3. We note that you are reporting an analysis of a microarray, next-generation sequencing, or deep sequencing data set. PLOS requires that authors comply with field-specific standards for preparation, recording, and deposition of data in repositories appropriate to their field. Please upload these data to a stable, public repository (such as ArrayExpress, Gene Expression Omnibus (GEO), DNA Data Bank of Japan (DDBJ), NCBI GenBank, NCBI Sequence Read Archive, or EMBL Nucleotide Sequence Database (ENA)). In your revised cover letter, please provide the relevant accession numbers that may be used to access these data. For a full list of recommended repositories, see http://journals.plos.org/plosone/s/data-availability#loc-omics or http://journals.plos.org/plosone/s/data-availability#loc-sequencing.

Reviewers' comments:

Reviewer's Responses to Questions

**Comments to the Author**

1. Is the manuscript technically sound, and do the data support the conclusions?

Reviewer #1: Partly

Reviewer #2: Yes

2. Has the statistical analysis been performed appropriately and rigorously? 

Reviewer #1: I Don't Know

Reviewer #2: Yes

3. Have the authors made all data underlying the findings in their manuscript fully available?

Reviewer #1: No

Reviewer #2: Yes

4. Is the manuscript presented in an intelligible fashion and written in standard English?

Reviewer #1: Yes

Reviewer #2: Yes

5. Review Comments to the Author

Reviewer #1: Summary:

Herrmann et al present a manuscript describing the phenotype the Fetuin-A (Ahsg) knock out mouse on the DBA/2 background (D2,Ahsg-ko) using microCT, electron microscopy, transcriptional and proteomic analysis. Major finding stated are the reduced growth and premature death of these mice as compared to strain-matched controls due to extensive ectopic calcification of the soft tissues. The authors conclude that calcification in these mice initiates in the lumen microvasculature due to the absence of the fetuin-A mediated solubilization and clearance of excess calcium-phosphate from the circulation.

Comments:

It appears that females are strikingly more sensitive to an absence of Fetuin-A than their male counterparts, but this was not addressed at any length, other than mentioning this difference. Do WT females have more circulating feutin-A at baseline? Are women more prove to CKD-related calcification than men? Do they have less fetuin-A? Was the cardiac output shown in S1 in males or females? This information would be more informative if output was also graphed by gender. The cause of death was stated as “most likely” due to cardiac failure, yet, there was no histology (for example, picrosirius red, masson trichrome) supporting this. Additionally, as the pancreas was severely calcified, how is the glucose tolerance of these mice?

The authors make the strong statement that calcification is not associated with osteogenic differentiation. This statement is based on genome wide expression analysis of the kidney and brown fat and the data does not show increased expression of traditional makers of osteogenic differentiation. This conclusion is an overstatement. The several pre-ceding figures clearly shows that calcification initiates in the microvasculature, yet this transcriptional analysis was performed on whole organ tissue, not on microvascular cells. The precise cells involved in microvascular mineralization (ECs, SMCs) should be used for this sort of analysis. In the absence of collecting these cells in vivo, lines from WT and D2,Ahsg-ko cells could be made. This statement “seems that osteochondral cell differentiation is consequence, rather than a cause, of mineralization in soft tissues“, appears to apply this argument to all forms of ectopic calcification; this is not at all supported by the data in this study and should be rephrased.

Minor Comments:

The symbols in Fig1 are difficult to discern. They should be made larger or color coded.

The proteomic data in table1-3 could be better organized/grouped together, and more importantly, what are some of the highly up/down regulated proteins?

The data from the gene expression array and the proteomic analysis must be put on a publicly available server, per PLOS ONE rules.

Reviewer #2: This is a really important manuscript for the study of cardiovascular calcification in general. The manuscript presents an animal model that is prone to vascular calcification and evaluate the origins and characteristics of the protein-mineral particles formed in the vasculature and their relation to Fetuin-A.

As said, this is an important addition to the subject, but during the discussion of the manuscript the text ignores several works on the literature that indeed support several of the claims presented in the work.

Cardiovascular calcification has been extensively characterised in the literature and for example the work of Bertazzo have demonstrated that the calcification present in the vascular tissue present different physico-chemical characteristics to bone, what supports the conclusions presented in this manuscript. Moreover, there is also other works by Hutcheson, that also present characterisation and studies about possible origin for vascular calcification that is similar to the calcification presented by the authors, again supporting several of their conclusions.

Therefore, the results presented in the mentioned works should also be discussed in the manuscript what would help readers to have a better idea of the existing (and sometimes competing) models and ideas about vascular calcification.

Finally, would be good to have some scanning electron micrographs of the calcification and not only transmission micrographs, since the SEM would present a good idea of the 3D morphology of the calcification and even would help to understand the similarities and differences between the calcification presented by the authors and previous characterisation of human vascular calcification.

6. PLOS authors have the option to publish the peer review history of their article (what does this mean?). If published, this will include your full peer review and any attached files.

Reviewer #1: No

Reviewer #2: No

---

## [Author Response · Author response to Decision Letter 0]

23 Dec 2019

please see the file "response to the reviewers".

---

## [Decision Letter · Decision Letter 1]

17 Jan 2020

Lumenal Calcification and Microvasculopathy in Fetuin-A-Deficient Mice Lead to Multiple Organ Morbidity

PONE-D-19-27423R1

Dear Dr. Jahnen-Dechent,

We are pleased to inform you that your manuscript has been judged scientifically suitable for publication and will be formally accepted for publication once it complies with all outstanding technical requirements.

With kind regards,

Elena Aikawa, MD PhD

Academic Editor

PLOS ONE

Additional Editor Comments (optional):

Reviewers' comments:

Reviewer's Responses to Questions

**Comments to the Author**

1. If the authors have adequately addressed your comments raised in a previous round of review and you feel that this manuscript is now acceptable for publication, you may indicate that here to bypass the “Comments to the Author” section, enter your conflict of interest statement in the “Confidential to Editor” section, and submit your "Accept" recommendation.

Reviewer #1: All comments have been addressed

Reviewer #2: All comments have been addressed

2. Is the manuscript technically sound, and do the data support the conclusions?

Reviewer #1: Yes

Reviewer #2: Yes

3. Has the statistical analysis been performed appropriately and rigorously? 

Reviewer #1: Yes

Reviewer #2: Yes

4. Have the authors made all data underlying the findings in their manuscript fully available?

Reviewer #1: Yes

Reviewer #2: Yes

5. Is the manuscript presented in an intelligible fashion and written in standard English?

Reviewer #1: Yes

Reviewer #2: Yes

6. Review Comments to the Author

Reviewer #1: All of my previous comments were addressed by the authors of this manuscript and I feel that it is acceptable for publication

Reviewer #2: I am happy with the authors answers and modifications done to the manuscript. Therefore I would recommend this work for publication.

7. PLOS authors have the option to publish the peer review history of their article (what does this mean?). If published, this will include your full peer review and any attached files.

Reviewer #1: No

Reviewer #2: No

---

## [Editor Report · Acceptance letter]

22 Jan 2020

PONE-D-19-27423R1 

Lumenal Calcification and Microvasculopathy in Fetuin-A-Deficient Mice Lead to Multiple Organ Morbidity 

Dear Dr. Jahnen-Dechent:

I am pleased to inform you that your manuscript has been deemed suitable for publication in PLOS ONE. Congratulations! Your manuscript is now with our production department. 

With kind regards,

on behalf of

Dr. Elena Aikawa 

Academic Editor

PLOS ONE